# Deep tensor factorization models of first impressions

**Yangyang Yu & Jordan W. Suchow**
School of Business
Stevens Institute of Technology
Hoboken, NJ 07030
`{yyu44,jws}@stevens.edu`

## Abstract

Machine-vision representations of faces can be aligned to people's first impressions of others (e.g., perceived trustworthiness) to create highly predictive models of biases in social perception. Here, we use deep tensor fusion to create a unified model of first impressions that combines information from three channels: (1) visual information from pretrained machine-vision models, (2) linguistic information from pretrained language models, and (3) demographic information from self-reported demographic variables. We test the ability of the model to generalize to held-out faces, traits, and participants and measure its fidelity to a large dataset of people's first impressions of others.

## 1. Introduction

People form first impressions of others based in part on their appearance [1]. Computational models of person perception often focus on modeling the face features that give rise to those first impressions [14, 17], the semantic structure of trait space and the relationships between traits [7], or individual- and group-level differences in person perception [10].

To understand the visual basis of people's biased first impressions, researchers have aligned representations from machine-vision systems to large datasets of first-impression judgments [8]. However, people's first impressions depend not only on the visual information present in an image of a face, but also on the interaction between (1) observers, (2) the faces of the people they encounter, and (3) the traits along which the observers form impressions. Thus, a complete computational model of first impressions must also reflect representations of traits and individual- and cultural-level differences across observers.

Recent advances in machine-learning methods for multimodal fusion provide a means to create unified representations of heterogeneous data [4]. Here, we developed an integrated approach based on deep tensor factorization with side information [9] that aligns representations from multiple pretrained deep networks (dense vector representations of images of human faces, textual descriptions of psychological traits, and demographic features of participants) to create predictive models of first impressions. This method transforms each machine representation into a modality-specific latent space to combine with the other modes. For behavioral data, it embeds information from the different channels into separate latent spaces formed by deep feature vectors, then uses an attention mechanism to interpret between-channel interactions.

In this paper, we first describe the structure of our deep tensor-factorization model. We then train the model using a large-scale dataset of first impressions and finally assess the ability of the model to generalize to held-out faces, traits, and participants.

Preprint. Under review at SVRHM.

## 2. Background And Related Work

Information from at least three channels contribute to people's first impressions of others: (1) visual information from faces, (2) linguistic or conceptual information about traits, and (3) demographic features of participants. Deep tensor factorization with side information is a mathematical framework for aligning information from multiple channels: it combines (through, for example, a dot product) pretrained latent vectors of features from different channels of the perceptual tensor into a single scalar value – one entry of the main data tensor. Moreover, it has the advantage of allowing additional channels to be incorporated and for more complex transformations to be used in service of combining the latent representations (e.g., attention mechanisms). We explain the development of the latent feature spaces for the three channels in detail below.

### 2.1. Modality 1: Visual Information about Faces

Valentine et al. (2016) [15] proposed a "face space" as a concept for representing psychological similarities of human faces. Each face can be represented by a point placed at a particular location in the face space. Distances between faces reflect their similarities. With developments in computer vision, algorithms can learn more comprehensive face features. The extracted face features may derive from interpretable attributes (e.g., chin width)[13], or features learned from various machine learning models (e.g., CNN and Eigenfaces [3, 14]) or deep learning algorithms [5, 6, 12]. The deep face features have much higher dimensionality and are generally more expressive.

### 2.2. Modality 2: Linguistic Information about Traits

The concept of trait space is often found in studies of person perception and first impressions [7, 11], which used dimensionality reduction and related techniques to explore the similarity structure of traits. However, their method for generating a trait space is limited in its ability to generalize to new traits that do not appear in the training dataset. Pretrained language models such as BERT [2] provide a possible alternative. BERT is known for extracting high-quality language features from text data and can create high-dimensional representations of any given input text. In the case of first-impression ratings [8], BERT can be applied to a compact set of texts with close semantic meaning for the given traits to generate latent vectors. These latent vectors form the input to the multidimensional trait space.

### 2.3. Modality 3: Demographic Information about Participants

Shaver et al. (2015) [10] distinguish between individual and cultural/ group differences in person perception. Here, we consider both to be participant side features. The one-hot-encoding method is applied to demographic features self-reported by participants to formulate the input to the demographic latent space.

## 3. Methods

We explored five deep tensor factorization models. The workflow in Figure. 2 (Appendix. A) shows their three major components: pretrained deep networks for feature latent space generation in each channel, a multichannel fusion strategy with attention, and dense layers for converting fused vectors into impression-rating predictions. Differences across the five models are mostly in the fusion strategy. Model 1 is vanilla tensor factorization, which takes the dot product of dimensionally reduced 3-way side feature vectors to generate impression-rating predictions. Models 2 to 5 are four deep tensor factorization approaches. Model 2, the additive concatenation approach, combines the 3 modes of side features into a single row vector. And Models 3, 4, and 5 apply a between-channel attention mechanism on top of the concatenated vector in Model 2.

Figure.1 describes the working mechanisms of all deep fusion models in detail. At first, the pretrained deep networks (described in Section 2) process the raw perceptual data for each channel in parallel and produce three sets of feature vectors. Then, different ways of information consolidation are conducted. In Models 1 and 2, information from all channels is merged without attention, while Models 3, 4 and 5, provide alternative methods for computing between-channel interactions. Models 3 and 4 are the two well-known attention mechanisms initially introduced

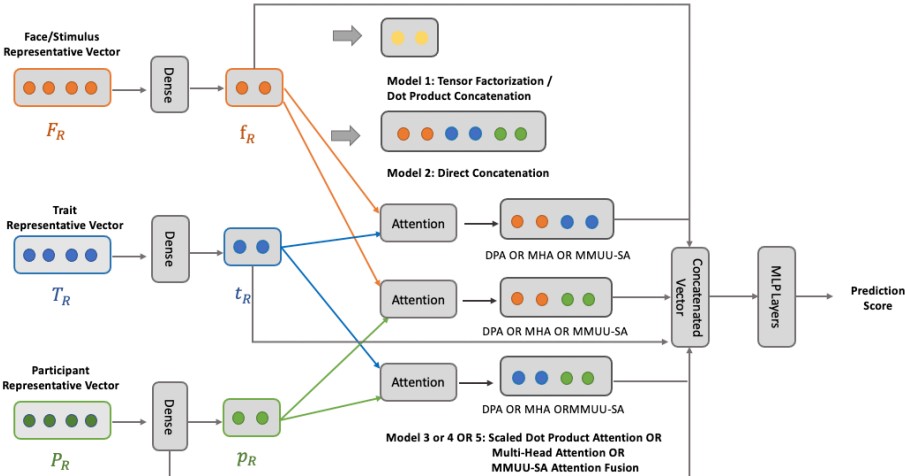

Figure 1: Structures of five options multi-modal fusion methods handling human face perception prediction data.

in [16] (details seen in Appendix.B and D). Model 3 uses Scaled Dot Product Attention (DPA), which is applied to every pair of the three channels. Model 4 uses Multi-Head Attention (MHA), which has a more complex way of computing attention. It calculates attention from multiple heads by repeatedly querying the data. Option 5 is inspired by the Multi-Modal Uni-Utterance- Self Attention (MMUU-SA) in [4] (details seen in Appendix.E). All the calculated attention vectors are concatenated with the original vector representations of these three channels. By this means, a long fusion vector representing both within-channel and between-channel information is generated.

The final portion of the fusion models is a series of dense layers with non-linear transformations with flexible options such as dropout, batch normalization, and so on. Fusion vectors are fed into the dense layers to produce impression predictions. In sum, the pretrained networks produce vector representations for each channel, the tensor factorization fusion strategies decide how to integrate data from all channels together, and the dense layers produce the final output. Together, they work as an end-to-end system for predicting first impressions from the output of pretrained machine vision and language networks.

## 4. Results

To evaluate the performance of the tensor factorization approaches, we tested each model on the One Million Impressions dataset [8], which contains over one million first-impression ratings collected for over a thousand StyleGAN2-generated synthetic face images with respect to thirty-four distinct traits.

We conducted two sets of experiments to test the generalizability of the five fusion models. The first set compares their ability to predict first impressions at the individual level. Linear regression (Model 0) served as the baseline model. 95% of the dataset was used as a training dataset and 5% as the validation set. Under a learning rate ($lr = 0.002$), performance metrics for each model were collected after 100 epochs and reported using the average performance among three repeated experiments. The performance of each model on the validation set is summarized in Table. 1. All five tensor-factorization models (Models 1–5) outperform the baseline regression model considerably. Model 4, MHA with 8 heads, achieves the lowest RMSE (20.2538) and Model 5, MMUU-SA, achieves the best linear correlation ($R^2 = 0.4867$). Though each model has its own advantages, in training over epochs (Figure.3 in Appendix.C), the latter shows noticeably more distance between training and validation RMSEs in later epochs, implying a greater possibility of overfitting. As such, the 8-head MHA provides the best predictive model for individual-level first impressions. In contrast, Model 1, vanilla tensor factorization, performed the worst among all the tensor-factorization models.

Table 1: Performance metrics of deep tensor factorization models

| Model | Description | Key Parameters | RMSE | R-Square |
|-------|-------------|----------------|------|----------|
| 0 | Linear Regression | 512 face features + 300 trait features + 27 participant features, linear combination | 26.4936 | 0.2312 |
| 1 | Direct Dot Product | 512 face features, 300 trait features, 27 participant features all dense into 100 dimensions, calculate the dot product for the three 100-dimensional vectors | 21.0805 | 0.3073 |
| 2 | Direct Additive Concatenation | 512 face features, 300 trait features, 27 participant features all dense into 100 dimensions, concatenated into a 300-dimensional vector | 20.3380 | 0.4838 |
| 3 | DPA | Scaled Dot Product Attention + 300-dimensional vector generated by three-channel features concatenation | 20.4989 | 0.4805 |
| 4 | MHA | 3-head Attention + 300-dimensional vector generated by three-channel features concatenation | 20.2944 | 0.4805 |
|   |   | 8-head Attention + 300-dimensional vector generated by three-channel features concatenation | 20.2538 | 0.4761 |
| 5 | MMUU-SA | Multi-Modal Uni-Utterance- Self Attention + 300-dimensional vector generated by three-channel features concatenation | 20.2855 | 0.4867 |

The second set of experiments explored the vanilla tensor factorization approach in terms of its ability to predict first impressions of held-out entities, which might either be faces, traits, or participants. Vanilla tensor factorization was selected to conduct the experiments, for Table. 1 indicates it has the threshold performance of all models for individual-level prediction. Correlation coefficients were measured as a target metric for held-out faces, traits, and participants to compare with individual-level impression rating prediction. As a baseline for the comparison target, the training process of the individual-level prediction used all the data, with information associated with all faces, traits, and participants. However, the other models had no exposure to held-out entities of a particular type. Results demonstrate that models tested on held-out entities are lower than that of a model tested on held-out individual ratings ($R^2 = 0.48$). Despite this, the tensor factorization approach gains substantially comparable performance for holding out faces ($R^2 = 0.42$) and participants ($R^2 = 0.45$) concerning the target model. Nevertheless, its performance for held-out traits ($R^2 = 0.046$) is low, though above chance. This is likely because held-out faces and participants have more underlying similarities with the remainder used as training samples. In particular, the traits are few in number and were intentionally selected to be different from each other. Therefore, the natural connections among traits are limited.

## 5. Discussion & Conclusion

The deep tensor-factorization approach that we propose aligns the latent space of two or more pretrained deep neural network representations to behavioral judgments made by people. The system generalizes to multiple entities and their relations. We showed that, when modeling a

Table 2: R-Square values for held-outs of three channels via the tensor factorization fusion model.

| Held out channel | Sampling Strategy | R-Square |
|---|---|---|
| (Ratings) | Use all samples, 5% data used to compute correlation values, the rest used as training samples | 0.48 |
| Faces/Stimulus | Use all samples, the correlation coefficient of 100 faces computed at one time, data of the rest faces used as training samples | 0.42 |
| Traits | Use all samples, the correlation coefficient of 1 trait computed at one time, data of the rest traits used as training samples | 0.046 |
| Participants | Use samples related to 2000 out of 4476 participants, the correlation coefficient of 200 participants computed at one time, data of the rest participants used as training samples | 0.45 |

large dataset of human first impressions [8], deep tensor factorization methods outperformed the baseline regression and vanilla tensor factorization (Table. 1).

More research is needed to understand which deep tensor factorization approach provides the most reliable method for representing between-channel interactions. Though we found that MHA achieved the best performance based on the average of repeated experiments, performance was comparable across all the deep-tensor factorization methods. Further corroboration is needed to demonstrate that 8-head MHA is the optimal attention mechanism for representing these datasets. Moreover, other attention-based mechanisms are possible.

We also note that the generalization performance of the tensor factorization approach for held-out traits prediction is weak (Table. 2). We suggest two potential improvements: first, collect a much denser sampling of traits than the 35 traits examined here. This is important given the relatively high dimensionality of the language models used (300) in comparison to the number of traits tested (35). Second, we might apply alternative attention mechanisms to better capture between-channel correlations and predict impressions for held-out traits.

Overall, we show that, as one of the state-of-the-art fusion models, deep tensor factorization provides the key to creating computational models of biased first impressions. Its 3D tensor structures enable the integration of heterogeneous information, commonly seen in behavioral datasets. It substantially improves the alignment of human impression judgments with machine-generated deep fusion representations. Though the most appropriate choice of pretrained networks and attention can be further explored, the deep tensor factorization models demonstrate the ability to generalize to held-out faces, traits, and participants.

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

## Appendix A.
## Data organizing structure of deep tensor factorization modeling human attribute inference

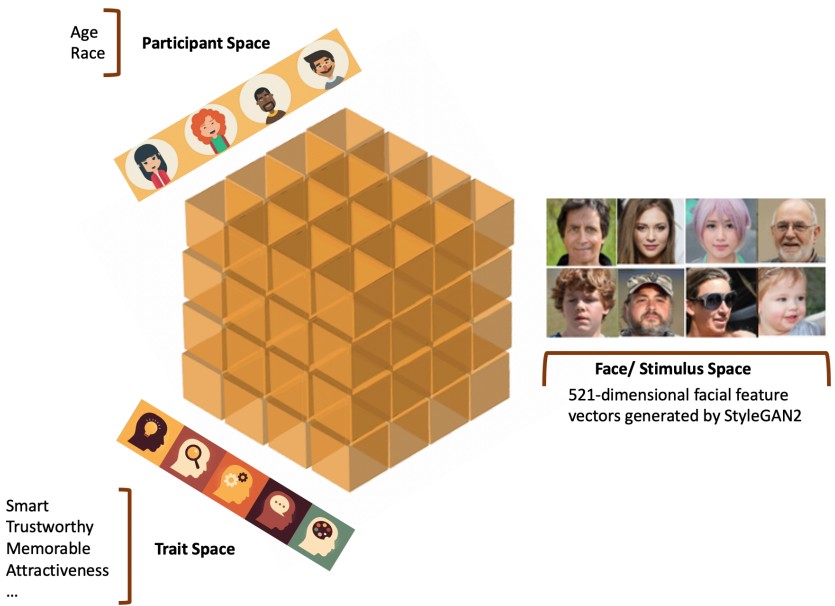

Figure 2: Data organizing structure of deep tensor factorization modeling human attribute inference. *Note: Each entry of the main 3D tensor in the middle of the graph stands for a face impression rating on a certain trait from one participant. The three side information are shown with detailed feature lists.*

## Appendix B.
## Scaled Dot Product Attention

According to the definition introduced by [16], an attention function can be described as mapping a query and a set of key-value pairs to an output, where the query, keys, values, and output are all vectors. In the face impression rating case, we can assign the query as the participant ($p_R$) as the query. keys are traits ($t_R$), and values are faces ($f_R$). All input vectors are dense vectors instead of the initial vectors. As one specific type of attention, Scaled Dot Product Attention (DPA) is computed as Formula. (1). The related working mechanisms can be briefly explained as follows: The $d_k$ denotes the total dimension number of inputs. $Q_{p_R}$ denotes a packed set of simultaneous queries, while $K_{t_R}$ and $V_{f_R}$ stand for its corresponding packed matrices of keys and values.

$$\text{Attention}(Q_{p_R}, K_{t_R}, V_{f_R}) = \text{softmax}(\frac{Q_{p_R} K_{t_R}^T}{\sqrt{d_k}}) V_{f_R} \tag{1}$$

## Appendix C.
## Training vs. validation RMSEs of deep tensor factorization models over 100 epochs

The plots in Figure. 3 below show the RMSE values over 100 epochs for five options of tensor factorization approaches. Comparing the difference in traces of training vs. validation data over epochs demonstrates 1) the potential overfitting issue for some approaches is more severe than others. As the overfitting issue affects the model's generalization ability, it can be used as a reference to distinguish whether the Multi-Head Attention Approach (MHA) is the optimal solution

for the face impression rating case over the Multi-Modal Uni-Utterance-Self Attention (MMUU-SA). 2) After a certain number of training epochs, the RMSEs in most of the approaches become relevantly stable and slowly decrease, except for the Scaled Dot Production Attention (DPA) approach having noticeable fluctuations.

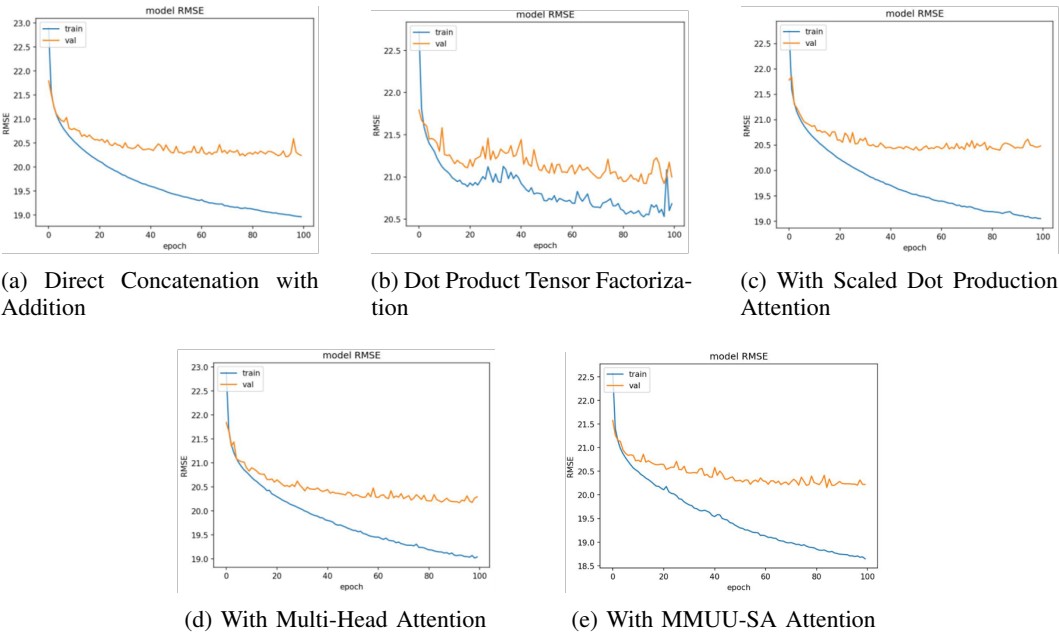

(a) Direct Concatenation with Addition

(b) Dot Product Tensor Factorization

(c) With Scaled Dot Production Attention

(d) With Multi-Head Attention

(e) With MMUU-SA Attention

Figure 3: Training vs. validation RMSEs of deep tensor factorization models over 100 epochs.

# Appendix D.
# Multi-Head Attention

In the Multi-Head Attention approach, queries, keys, and values are project $h$ times by distinct learned linear projections (to dimension $d_k$), where $h$ is the number of heads. This mechanism make the overall model able to jointly summarize information from different representation subspaces at different positions. In Formula. (2), $W_i^{Q_{p_R}}, W_i^{K_{t_R}}, W_i^{V_{f_R}}$ denote the parameter matrices of projections.

$$\text{MultiHead}(Q_{p_R}, K_{t_R}, V_{f_R}) = \text{Concat}(head_1, ..., head_h)W^O$$
$$\text{where } head_i = \text{Attention}(Q_{p_R}W_i^{Q_{p_R}}, K_{t_R}W_i^{K_{t_R}}, V_{f_R}W_i^{V_{f_R}}). \tag{2}$$

# Appendix E.
# Multi-Modal Uni-Utterance-Self Attention

In the Multi-Modal Uni-Utterance-Self Attention approach, Formula. (3), (4) shows its basic computational operations. The $X \in \mathbb{R}^{3 \times d}$ in Equation. (3) stands for the information tensor, where the three d dimensional rows are the outputs of the dense layer ($p_R, t_R, f_R$ in Figure. 1) for the three modalities. The attention tensor $A \in \mathbb{R}^{3 \times d}$ is computed in Equation.(4), concatenated with $X_{p_R}$, and passed to the output impression predicting layer.

$$M = X.X^T. \tag{3}$$

$$N(i,j) = \frac{e^{M(i,j)}}{\sum_3^{k=1} e^{M(i,k)}} \ for \ i,j = 1,2,3$$
$$O = N.X$$
$$A = O \odot X. \tag{4}$$