# OpenReview forum: "Deep tensor factorization models of first impressions"
_NeurIPS.cc/2022/Workshop/SVRHM — SVRHM Poster_

### Official Review · Reviewer_FBA9 · 2022-10-07
**Novel approach on an important topic, but motivation and technical details could be clarified**

**Rating:** 8
**Confidence:** 4

**Review:**

This is an interesting paper using deep tensor factorization models to understand human judgements of first impressions. The authors compare different tensor factorization techniques to combine 1) visual information about faces, 2) trait (semantic) information, and 3) participant demographic information.

While I liked the paper and approach, I was confused about some of the motivation and several technical details.

It wasn't clear to me why the authors decided to include linguistic/semantic information about traits. Face trait judgements are made within a few milliseconds. Do the authors think humans engage similar semantic reasoning on this timescale? (In fact, this seems to have very low prediction in experiment 2.)

I understand space was tight but the methods text was somewhat confusing. The term ‘3-way side feature vectors’ is introduced without explanation in the methods. I believe that refers to the 3 modalities described above, but that should be clarified, and it would help to define ‘side features’ for those unfamiliar. It would also help the reader to add Model #’s to the appendix.

A smaller point is that the names of the 3 modalities should be consistent across the paper. I think 'visual' and 'semantic' would be more clear than 'face' and 'trait' which could both be visual or semantic.

The text just says the baseline model is a linear regression. A regression between what? All 3 types of features and impressions? This seems like an unfair comparison given the number of computational layers in models 1-5.

A more interesting question would be to separate out the 3 types of information to understand the contribution of each. I think this is what Experiment 2 aimed to do, but I did not understand these. Didn't all models predict participant trait judgements? How were the different entities held out and then predicted?

---

> ### Author Response · Authors · 2022-11-24
> **Thanks for reviewing our paper and a few points in response to the confusing points**
>
> We are very happy you like this research idea. And there are some further explanations about the points you mentioned in your review. Hopefully, it will resolve your confusion about our research.
>
> 1) We agree with your opinion that people's response time could also be an important factor to consider, and thanks a lot for bringing it up. And luckily, the One Million Impressions dataset does contain this variable. We are planning to include it in future experiments as well. And we are thinking about what is the best way to make use of it to let it show the maximum effects in impression prediction.
>
>  2) We consider low prediction for held-out is primarily caused by the much smaller trait set than sets of other channels (e.g., faces). We only have 35 traits, and most of them have very distinctive meanings from each other. The trait space they construct is relatively sparse, making it harder for the algorithm to learn people's impression tendencies compared to cases with many more traits. This point can be supported by the face space with a much denser form. The prediction result for held-out faces turned out to be much better than that of traits. So moving forward to validate our idea further, rating for more types of traits needs to be collected. As our model makes the most impact when learning the interactions between objects, we also plan to select traits with similar and opposite meanings to the existing traits for the model to learn.
>
> 3) The linear regression is based on all the face, trait, and demographic features. We use it because it can be most of the fundamental baseline in the multi-modal learning area. At the same time, it aligns with the methodology shown good results in previous research done with the same public dataset (One Million Impressions dataset). So we think it can be a good benchmark.
>
> 4) In response to the questions, 'Didn't all models predict participant trait judgments? How were the different entities held out and then predicted?' you proposed for Experiment 2, first, Model 1 in the Table. 1 -- Direct Dot Product model is used in experiment 2. We chose it because it beat the baseline model of linear regression. At the same time, it showed the poorest performance among all the tensor factorization models in terms of RMSE and R-Square when predicting individuals' impression ratings. So if its ability to predict held-out channels (Faces/ Participants) can be proven, we can hold even more optimistic expectations for the rest of the tensor factorization models. Because compared to holding out random data points as a test set in individual-level prediction (the first experiment), the second experiment can be viewed as a specific case on pre-defined data points as the test set. Second, we use the face channel as an example to clarify how to conduct the held-out experiment. Say we have 1000 face images, every time we hold out all the ratings and their related features for 10 of them as test data and use the rest to predict the impression ratings for the held-out 10 faces.

---

### Official Review · Reviewer_7tig · 2022-10-14
**Very interesting work combining various modalities towards predicting first impressions**

**Rating:** 9
**Confidence:** 5

**Review:**

The paper was fairly clear to follow. Please see the couple of suggestions below. Overall this work would be very interesting for the audience of SVRHM. Multi-modality is intrinsic to the real world. This work presents some fascinating comparisons of various ways to combine this plethora of information towards people first impressions. What would be super cool as future work on this direction is also combining audio data as yet another source of features. Aside from visual information and language used, I could see a lot of bias towards first impressions coming from audio input (aka how people sound).

nit: Please consider changing Model 1, 2,...5 to Option 1, 2,..., 5 in the text or vice versa (change Option 1, 2,..., 5 to Model 1, 2,..., 5 in the figure) to match the two. It took a bit of starring to make the right connections between text and figure.

"Vanilla tensor factorization was selected to conduct the experiments, for Table. 1": Vanilla tensor factorization is mentioned as Model 1 from Models 1-5 the paper compares. Table 1 shows results from all 5 models including the linear regression baseline. Could you please clarify what this sentence means?

---

> ### Author Response · Authors · 2022-11-24
> **Thanks for reviewing our paper and changes have been make based on your helpful suggestions.**
>
> We are so happy you like this research idea, and thanks for providing us with constructive suggestions.
>
> We have made the change to the notations related to the five models in the camera-ready version to make the information consistent.
>
> In response to the question about vanilla tensor factorization you proposed in the review, it points to Model 1 in the Table. 1 -- Direct Dot Product model. We chose it because it beat the baseline model of linear regression. At the same time, it showed the poorest performance among all the tensor factorization models in terms of RMSE and R-Square when predicting individuals' impression ratings. So if its ability to predict held-out channels (Faces/ Participants) can be proven, we can hold even more optimistic expectations for the rest of the tensor factorization models. Because compared to holding out random data points as a test set in individual-level prediction (the first experiment), the second experiment can be viewed as a specific case on pre-defined data points as the test set.

---

### Official Review · Reviewer_pUcB · 2022-10-14
**Limited motivation, experimental setup and conclusions, among other aspects**

**Rating:** 3
**Confidence:** 5

**Review:**

This papers presents a set of machine learning models to predict people's first impressions of others, merging images of faces, linguistic information from a language model and demographics, via deep tensor factorisation.

My overall impression of this submission is that there are multiple aspects in which the paper could be improved, hence my rating below acceptance level.

* Motivation: While I understand the goal of the paper, I miss a solid motivation of why it is important to develop machine models of people's first impressions of others and what is the research gap that this paper is aiming to fill.
* Ethics: Related to motivation, I would also expect a discussion about the ethical implications of this work, given that it machine models of people's first impressions could potentially acquire biases harmful to some groups or individuals, as well as being misused.
* Experimental setup: The paper studies the performance of five models as well as a baseline. First, I missed a discussion of the motivation for the choice of methods. Why is deep tensor factorisation a method of interest for this problem? Why are simpler methods not studied? Second, I would argue that the baseline used (linear regression) is significantly weaker than the deep tensor factorisation methods. In between linear regression and deep tensor factorisation there exists a multitude of methods that could be more appropriate baselines to judge the performance of the proposed models.
* Conclusions: In my opinion, the paper falls short of providing an in-depth analysis and conclusion from the work presented in the paper. The main conclusion by the authors seems to be that "deep tensor factorization methods outperformed the baseline regression and vanilla tensor factorization". What are the implications of this conclusion and how does it advance our knowledge about what? I believe the discussion remains at a quite superficial level. Later, one paragraph reads that the paper shows that "deep tensor factorization provides the key to creating computational models of biased first impressions", but I disagree that the paper shows such a strong statement. Nevertheless, the discussion is also shallow in this regard.
* Writing: The writing of the paper could also be polished. For instance the article uses informal expressions ("and so on"), contains non rigorous statements ("various machine learning models (e.g., CNN and Eigenfaces [3, 14]) or deep learning algorithms"; "Under a learning rate..."), and some paragraphs that are not easy to read, such as the last paragraph of Section 4. As a more minor comment, I would not use a "." after "Figure" or "Table" since the whole word is used.

---

> ### Author Response · Authors · 2022-11-25
> **Thanks for your review and suggestions.**
>
> Thanks a lot for your review. Here are a few further explanations in response to what you proposed in the comment. Hopefully, they can answer your questions about our paper.
>
> Motivation  & Ethics:  The main research gaps to be filled are: 1) Given features clearly existing multi-layer correlation, what is the better way to use it fully and to untangle the intricacies of people's impression judgments of human faces? 2) Is there any appropriate and generalized framework used to numerically represent the sophisticated interactions commonly seen on many social science datasets at a large scale?
> The potential need to align human judgments and decision-making with machine algorithms in practice. It helps people learn the human impression-formulating process, which they themselves may barely be able to notice. It provides a novel perspective for human cognition in the general understanding of individual behavior. At the same time, it helps develop more intelligent machine systems. Notice that, by using face images generated by machine algorithms, this research is free of the ethical concern of making judgments on real humans.
>
> Experimental setup:  There seems to be a concern about the choice of method. We chose linear regression as a baseline because there are previous researches using linear regression on the same dataset showing good results on impression prediction. So it is used as a comparison benchmark for our research. Of course, we are open and welcome to other methods if you want to propose potential options. But we do believe that the deep tensor factorization method has its rationality and advantages to handle such large-scale social science datasets. It provides various ways to represent the different kinds of interaction and dependencies within data in an automatic learning process. Moreover, the flexible structure of our model makes it easy to switch to the most appropriate attention mechanisms to capture different layers of attention in the data. This unique structure also substantially improves the model's generalizing ability.
>
> Conclusions:  We understand your point that we could provide a deeper analysis and explain more about the detailed implications of the proposed model in our paper. But there is a limitation on page numbers, and we prioritized the most important information shown in the main content. If you would like to discuss any points further, please feel free to reach out.
>
> Writing:  Thanks again for your suggestion. We will pay more attention to the wording.

---

### Official Review · Reviewer_9kdB · 2022-10-17
**Interesting, but some methodical concerns**

**Rating:** 6
**Confidence:** 3

**Review:**

The paper discusses various techniques to factorize a 3D tensor of first
impressions, where the different modalities are face stimuli, traits, and
participants.

The objective of the paper is interesting, but I have some methodical concerns.

Mainly, the authors don't seem to take into account the number of parameters
that each of the tested methods has. Wouldn't it be sensible to include some
form of Occam's razor here, and not purely rely on cross validation to ensure
the models aren't overfitting?

In addition, the selection of models that are compared is somewhat biased
towards attention. Model number 2 (direct concatenation) doesn't perform that
badly, so perhaps giving the final MLP a few more layers would have been
sufficient to reach a similar performance?

Overall, I would also liked to have seen whether there are any qualitative conclusions that could be drawn from the resulting factorizations. I.e. what do we learn from this experiment about first impressions?

Other than this, some notes regarding clarity:

- It wasn't immediately clear to me what the loss function for training the
  "deep" tensor factorization methods is, and that the authors used cross
  validation with a held-out dataset to arrive at the results. This could be
  presented earlier on in the paper.

- In Figure 1, "Option 2" does not connect to the prediction score. This seems
  misleading, given that from the rest of the paper, it seems that that method
  still relies on the final MLP for making a prediction.

- It is misleading to talk about an "optimal" model for representing the
  dataset. Optimal means that there provably can't be a better model. I don't think that
  claim can be substantiated here.

---

> ### Author Response · Authors · 2022-11-24
> **Thanks a lot for reviewing our paper and a few explanation about the proposed questions**
>
> Thanks for proposing several valuable questions about our research. In response, here are explanations about our methodologies and experimental setup choices.
>
> 1) There seems to be a concern from you about why we chose the high-dimensional pre-trained feature sets instead of the lower ones. In fact, the feature extraction means of StyleGAN for images and BERT for texts have already been tested to help learn more subtle similarities and differences among image and text entities through many practical tasks in areas of computer vision and natural language processing. This point is also supported by one of the previous works done by Dr. Peterson et al. in our citations (citation number 8). In his research, it reached a promising result in people's impression rating prediction using solely facial features extracted from StyleGAN. The good result is mostly brought by its ability to represent human facial features in a very comprehensive manner. In the opposite of overcomplicating the problem, the implementation of deep learning provides a probability of automatically and efficiently generating a reliable feature set for impression prediction purposes. At the same time, it helps avoid biases of human judgments in the face feature defining and selecting process. Therefore, we extended this idea to use the deep learning pre-trained model to generate trait features as well.
>
> 2) Second is the explanation in response to the complexity concern of our proposed method -- deep tensor factorization itself. We use it mainly because we aim to solve the interactions within and between different entities (faces, traits, and participants) with a smart and efficient solution. Compared to the traditional model, like the linear regression model, it provides various ways to represent the different kinds of interaction and dependencies within data in an automatic learning process. Moreover, the flexible structure of our model makes it easy to switch to the most appropriate attention mechanisms to capture different layers of attention in the data. This unique structure also substantially improves the model's generalizing ability of other social science datasets. On the contrary, in linear regression, you may need to consider a bunch of pre-assumptions to be satisfied, conduct correlation analysis among variables, and repeat to test what kind of transformations and interaction terms to be added. These issues are very time-consuming and lose the theoretical ground to support the final result, even if only minor parts are violated, which is more likely to happen in high-dimensional data cases like ours. So, we consider that the development of our model fits the potential analytical needs of this face impression rating dataset as well as other social science datasets at large scale or high dimensions.
>
> 3) In response to your mention that 'perhaps giving the final MLP a few more layers would have been sufficient to reach a similar performance,' we totally agree and think it is a reasonable approach.
>
> We actually did it already. In our experiment, we chose the different layers to develop the model and only reported the best-performed one.
>
> Hope that the above answers all your concerns. If there are any other questions, please feel free to reach out!